# Morphological and Molecular Analysis of Australian Earwigs (Dermaptera) Points to Unique Species and Regional Endemism in the Anisolabididae Family

**DOI:** 10.3390/insects10030072

**Published:** 2019-03-14

**Authors:** Oliver P. Stuart, Matthew Binns, Paul A. Umina, Joanne Holloway, Dustin Severtson, Michael Nash, Thomas Heddle, Maarten van Helden, Ary A. Hoffmann

**Affiliations:** 1School of BioSciences, Bio21 Molecular Science and Biotechnology Institute, The University of Melbourne, Parkville, Victoria 3052, Australia; oliver.stuart93@gmail.com (O.P.S.); matthew.binns@csiro.au (M.B.); pumina@unimelb.edu.au (P.A.U.); 2Agriculture and Food Business Unit, Commonwealth Scientific and Industrial Research Organisation, Black Mountain, Australian Capital Territory 2601, Australia; 3Cesar, 293 Royal Parade, Parkville, Victoria 3052, Australia; 4New South Wales Department of Primary Industries, Wagga Wagga Agricultural Institute, Pine Gully Road, Charles Sturt University, New South Wales 2795, Australia; joanne.holloway@dpi.nsw.gov.au; 5Department of Primary Industries and Regional Development, South Perth, Western Australia 6151, Australia; dustin.severtson@dpird.wa.gov.au; 6School of Agriculture, Food and Wine, the University of Adelaide, Urrbrae, South Australia 5064, Australia; whatbugsyou@gmail.com (M.N.); Maarten.VanHelden@sa.gov.au (M.v.H.); 7School of Life Science, College of Science, Health and Engineering, La Trobe University, Bundoora, Victoria 3086, Australia; 8South Australian Research and Development Institute, Entomology, Waite Road, Waite, Urrbrae, South Australia 5064, Australia; thomas.heddle@sa.gov.au

**Keywords:** dermaptera, earwigs, Anisolabididae, barcoding, phylogenetics

## Abstract

Dermaptera (earwigs) from the Anisolabididae family may be important for pest control but their taxonomy and status in Australia is poorly studied. Here we used taxonomic information to assess the diversity of southern Australian Anisolabididae and then applied *cox1* barcodes as well as additional gene fragments (mitochondrial and nuclear) to corroborate classification and assess the monophyly of the putative genera. Anisolabididae morphospecies fell into two genera, *Anisolabis* Fieber and *Gonolabis* Burr, based on paramere morphology. Combinations of paramere and forceps morphology distinguished seven morphospecies, which were further supported by morphometric analyses. The morphospecies were corroborated by barcode data; all showed within-species genetic distance < 4% and between-species genetic distance > 10%. Molecular phylogenies did not support monophyly of putative genera nor clades based on paramere shape, instead pointing to regional clades distinguishable by forceps morphology. This apparent endemism needs to be further tested by sampling of earwig diversity outside of agricultural production regions but points to a unique regional insect fauna potentially important in pest control.

## 1. Introduction

Dermaptera (earwigs) represent a cosmopolitan [1], but understudied, insect order, totalling roughly 1930 species worldwide [2]. Comprehensive lists of dermapteran fauna are few, the most recent text being Steinmann’s *World Catalogue of Dermaptera* [3] as well as Haas’ online database [4]. Steinmann also authored selected volumes of the monograph series, *Das Tierreich*, which are considered the most recent comprehensive taxonomic works for Dermaptera [5,6,7,8]. Cassis [9] has most recently reviewed the Australian Dermaptera. He accounts for 85 species, 36 genera, and 18 subfamilies representing 7 of the 9 families recognised by Haas [10], while Haas’ database lists 88 species [11]. The most speciose families in Australia are the Anisolabididae and Spongiphoridae (=Labiidae; [6]), of which many specimens remain undescribed [9]. Cassis [9] lists 20 species of Anisolabididae in six genera in Australia, while Haas [11] puts this count at 32 species in 10 genera. Aside from this, there are very few works available that catalogue Australian Dermaptera, although such works do exist [12] (p. 364). Further, the phylogeny of the order is obscure. Numerous conflicting family arrangements have been proposed [13]. The two epizoic taxa, Hemimerina and Arixeniina, were previously considered suborders [10,14,15] because of their distinctive superficial morphology and lifestyle, or were not considered at all [3,5]. These have been reconsidered as highly derived sister taxa to other extant families [16,17,18], in line with Popham’s [19] morphological analysis. Molecular phylogenetic studies have typically been of insufficient depth to resolve generic relationships but have made progress towards resolving the familial phylogeny, which has undergone several recent revisions [16,17,18]. Dermaptera have recently come to the attention of the Australian grains industry where they are regarded as both pest and beneficial insects. Reports of crop damage by *Forficula auricularia* L. (Forficulidae), the European earwig, have become common over the last two decades [20], and this species is listed as an economically significant pest in Australia [21]. The Australian native *Nala lividipes* Dufour (Labiduridae), the black field earwig, is regarded as a pest of grain sorghum [22]. *Euborellia annulipes* (Lucas), a cosmopolitan species found in Australia [9] is a known pest of cultivated plants in Europe [23], but its status in Australia is unknown. *Forficula auricularia* is also considered a predator of several orchard pest arthropods [24,25,26,27,28,29,30]; *N. lividipes* is also considered a predator [31] along with confamilials *Labidura riparia* Pallas [32] and *Labidura truncata* Kirby [33]. We treat *L. truncata* as a distinct species following Giles and Webb’s [34] finding that its karyotype differs from *L. riparia*, although *L. riparia* shows karyotypic variation (see [35] and references therein) and not all authors accept the name *L. truncata* [36]. In addition, many earwigs have never been identified to species and their taxonomic status and agricultural importance remain unknown.

Among the earwigs collected from southern Australian grains systems, some species including *F. auricularia*, *L. truncata*, and *N. lividipes* have been collected and can be identified morphologically, but morphological characterization of many adult specimens remains incomplete. The most easily distinguished structure of dermapteran anatomy is the forceps, representing sclerotised, non-segmented cerci that are curved and exaggerated in males and straight and reduced in females [14], which play a role in male-male competition and in male displays to females [37,38]. This sexual dimorphism, as well as intrasexual male polymorphism in some species [39,40], make it difficult to delineate morphospecies based on these structures, particularly in mixed sex samples. Male genital morphology, while less apparent, is another trait used for distinguishing Dermaptera from each other at all taxonomic levels [41,42], although for some closely related species, male genital morphology may be uniform despite superficial morphological differences [43]. These two traits are both male-specific traits, and so female morphology is often overlooked in dermapteran taxonomy.

While molecular data have helped in resolving dermapteran phylogeny [13,16,17,18], they have not been used much in dermapteran biodiversity research. While DNA barcoding involving short nucleotide fragments has generally provided an aid to species identification and delineation of cryptic species [44], the approach has been criticised for a variety of reasons [45,46,47,48,49] and barcodes may suffer from considerable Type I and II errors depending on taxonomic knowledge and sampling of a clade [50]. A barcode gap, a lack of overlap between intra- and interspecific genetic distance which is crucial to barcoding’s utility, is not always found for a given set of taxa [51], and gene trees estimated using barcodes may not agree with well-established phylogenies [51]. Nevertheless, barcoding is common practice in contemporary invertebrate systematics [52,53,54,55], and represents just one type of molecular tool that can be employed; molecular (particularly mtDNA) evidence in taxonomic, systematic, and biodiversity research predates ‘barcoding’ in the literature by several years [56].

In this study, we analysed earwig samples from grains production areas of southern Australia to assess species diversity based on morphological information and further corroborate this diversity with molecular data. We then used multiple markers, mitochondrial and nuclear, to develop a molecular phylogeny of a subset of dermapteran diversity. These analyses provide the basis for describing earwig diversity in grains systems by highlighting the importance of regional endemism in earwigs. They should help to underpin future studies on the pest and beneficial status of different earwig taxa.

## 2. Materials and Methods

### 2.1. Sampling, Identification, and Morphology

We obtained whole adult specimens from surveys of dermapteran fauna in Victoria, New South Wales (NSW), South Australia (SA), and Western Australia (WA). Surveys comprised two types: a bi-annual (summer and winter) sample of representative sites across each state to investigate spatial variation in species composition and abundance, and a monthly sample of a reduced set of sites to investigate temporal variation. The collections from which adult Anisolabididae were sourced occurred from July 2016 to October 2018. Field sites covered the breadth of the Australian grain belt, except Queensland (Appendix A and Appendix A). Surveys were conducted using cardboard roll traps (a method developed for *F. auricularia* [57]) and pitfall traps (diameter = 4 cm, depth = 10.8 cm) filled with 50 mL 100% propylene glycol, both left at sites for a period of seven days each time. Specimens were stored in 100% ethanol at −20 °C. Specimens from pitfall traps were washed thoroughly with 100% ethanol prior to storage.

Males were distinguished from females based on forceps morphology, then classified into 10 morphospecies. Three (*F. auricularia*, *L. truncata*, and *N. lividipes*) were easily identified, particularly as only one of the cryptic sister species of *F. auricularia* [58] occurs in Australia [59]. The remaining earwigs were members of the Anisolabididae family, based on the lack of wings in the adult stage [6,9]. Generic and specific assignments were based on combinations of shapes of the male forceps and genitalia (Figure 1) following [6]. Some morphospecies could be identified to species, while others only to genus, and they are designated numerically. The Anisolabididae morphospecies were *Anisolabis* sp. 1, *Anisolabis* sp. 2, *Gonolabis forcipata* Burr (=*Mongolabis forcipata* [60]), *Gonolabis* nr. *gilesi* Steinmann (=*Mongolabis* nr. *gilesi* [60]), *Gonolabis* sp. 1, *Gonolabis* sp. 2, and *Gonolabis* sp. 3. A single male *Anisolabis maritima* (Bonelli) was found in Western Australia (site #6; Appendix A and Appendix A). As this is a well-known and recognisable species, it was not included in subsequent molecular or morphometric analysis which aimed to clarify the phylogeny and test the ability of available resources to distinguish the unknown Anisolabididae. It should be noted that, thus far, the Anisolabididae have not received a comprehensive phylogenetic, numerical, or cladistic treatment, and so any identification must be deemed provisional. To this end, it is noted that Srivastava’s [60] revision of Anisolabididae systematics would likely give different generic classifications. Dermaptera is an order rife with nomenclatural problems, and so it is difficult to treat any one text as authoritative. Specimens were pooled across sites and collection dates by morphospecies for subsequent morphometric and molecular analysis.

To test if the characters used for distinguishing morphospecies could be linked to other forms of morphometric measures, linear measurements were taken for all males (Figure 1) to assess the length (o) and width (p) of both of the forceps, the distance between the points of articulation of the forceps and the ultimate dorsal tergite (q), and the length (r) and width (s) of the right paramere. An index of the asymmetry of the forceps was calculated as the sum of the absolute differences between the greater of the two forceps lengths (o) and widths (p), divided by (q). The ratio of the right paramere’s length to its width was calculated, as well as the ratio of the maximum forceps length to the maximum forceps width. Typically, forceps asymmetry is measured by the absolute difference between the distances from the point of articulation to the tips of the forceps [40,61] but due to differences in body size between morphospecies and in the dimension of asymmetry (i.e., asymmetry in length, width, or both), a measure incorporating the two was deemed more appropriate. The final variables used for PCA were the forceps asymmetry index, the paramere length-width ratio, the basal width of the forceps, and the forceps length-width ratio.

Twenty-four hours prior to any dissection, specimens were transferred to 70% ethanol to make them more pliable. Parameres were removed by gently lifting the penultimate abdominal sternite and pulling out the entire male genitalia by its attachment to the sperm duct with forceps, and then cutting at the site of attachment. Photos were taken of male forceps and parameres with an IS500 5.0-megapixel camera (Tuscen Photonics, Fuzhou, China) attached to a SMZ1500 stereomicroscope (Nikon, Tokyo, Japan) with a C-Mount Adapter 0.45X (Nikon, Tokyo, Japan) at 40× and 80× respectively, using ISCapture version 4.1.3 (Tuscen Photonics, Fuzhou, China). A scale bar was photographed at each magnification. To assist repeatability of measurements, the material (either the dissected genitalia or the whole body of the adult specimen) was aligned along its anteroposterior axis to a digital graticule. All photos were taken from the dorsal side. Following this, all specimens were returned to 100% ethanol. Measurements were then taken from the photos using Fiji [62] with the appropriate scale photo used to set the measurement scale (pixels mm^−1^). For length and width measurements (Figure 1o,p,r,s), a bounding box was drawn encompassing the total vertical and horizontal width of each of the forceps and the right paramere. For the forceps, both forceps were measured, and the largest was taken. A random subset of 30 individuals was remeasured twice in a random order to assess repeatability. Intraclass correlation coefficients were calculated for each measurement, and these ranged from 0.713 (length of the left forceps) to 0.999 (length of the right forceps). Principal component analysis was performed on the scaled and centred morphometric variables. Excluding individuals with missing measurements (damaged forceps or parameres) the morphometric dataset comprised 124 male Anisolabididae specimens.

Having morphospecies defined *a priori* permits a classification analysis. A discriminant function analysis was performed with the morphometric dataset to assess the efficiency of simple linear measurements for species assignments. A first test was performed to assign morphospecies to putative genus, and then to putative species. In the case of the *Gonolabis* morphospecies, analyses were run both including and excluding *G*. sp. 3 as this morphospecies was only represented by three individuals, to assess the effect of a small class size on classification accuracy. For each case, the analysis was performed using the whole dataset to assess variable importance, and then with jack-knife cross-validation to assess classification accuracy. Note that classification accuracy was not assessed using a pre-defined proportion of training and test observations, because of the small size of the datasets (total N = 124, N*_Anisoalbis_* = 30, N*_Gonolabis_* = 94 or 91 excluding *G*. sp. 3).

### 2.2. Barcoding

To corroborate the morphospecies assignment by male characters, sequences of a 658 bp fragment of the cytochrome *c* oxidase subunit 1 gene (*cox1*) were obtained from selected individuals of each morphospecies across the sites at which they were found. Known species (*F. auricularia*, *N. lividipes*, *L. truncata*) were barcoded first. A few individuals (1–4, depending on the abundance at a site) were sequenced from at least 3 populations in each state to assess the barcode gap for widely distributed species. While barcode gaps may hold in the face of broad distributions [63,64], it is unknown how geographically structured dermapteran populations are, and so local-scale barcode divergence may confound the utility of the method. The number of individuals of each Anisolabididae morphospecies barcoded was largely determined by their individual abundances and their successful amplification. After successful retrieval of barcode sequences for male specimens, sequences for unknown female specimens were obtained. Individual females were selected based on their co-occurrence with males of each morphospecies. Females that fell within the range of genetic distance for a given male morphospecies (assessed visually using a neighbour-joining tree of p-distances generated in MEGA X [65]) were used to define adult female morphology for the morphospecies. All remaining adult females in the sample were then identified

Total genomic DNA extractions were performed using Chelex resin (Bio-Rad, Hercules, CA, USA) [66]. One of the mesothoracic legs was removed from individuals using forceps, taking care not to damage the body in the process and flame-sterilising instruments with 100% ethanol between handling different individuals. Excised legs were placed in 1.7 mL tubes with 200 µL 5% Chelex in solution and two 3 mm diameter glass beads, and then shaken in a TissueLyser II mixer mill (Qiagen, Venlo, The Netherlands) at 25 Herz for two minutes, following which the mixer mill adaptor plates were reversed in orientation and the tubes shaken for another two minutes. Tubes were spun down in a desktop centrifuge briefly and 5 µL of Proteinase-K (Bioline, London, UK) was added directly to the surface of the Chelex. Tubes were incubated in a water bath at 65 °C for two hours, followed by 10 min at 90 °C to deactivate the Proteinase-K. Extractions were stored at −20 °C. Prior to PCR, tubes were spun down at 18,600 rcf (× g) for seven minutes in a D3024 high-speed microcentrifuge (DLAB Scientific, Beijing, China), and aqueous DNA was pipetted from just above the surface of the Chelex.

We first used the “universal” arthropod primer pair LCO1490/HCO2198 [67] and where these failed used the revised degenerate versions jgLCO1490/jgHCO2198 [68]. DNA was amplified using 1 µL of total genomic extractions as templates in a 40 µL reaction with 1× Thermopol PCR buffer (New England Biolabs, Ipswich, MA, USA), 2 mM MgCl_2_, 200 µM dNTPs, 1 µM of each primer, 0.64 µg bovine serum albumin, and 1.6 U of *taq* polymerase (New England Biolabs, Ipswich, MA, USA). Initial denaturation took place at 95 °C for four minutes, followed by 40 cycles of 95 °C denaturation for 30 s, 49 °C annealing for 60 s, and 72 °C extension for 90 s, followed by 72 °C for three minutes for the final extension using a Mastercycler EP Gradient S thermocycler (Eppendorf, Hamburg, Germany). Products were run on a 2% agarose gel set with SYBR Safe DNA Gel Stain (ThermoFisher, Waltham, MA, USA) at 100 eV for 40 min, and then visualised with GelDoc XR+ (Bio-Rad, Hercules, CA, USA) using the default setting for SYBR stained DNA. Products were directly sequenced in both directions using the respective primers with an Applied Biosystems 3730 capillary analyser (Macrogen, South Korea). and others: sequence data were generated with Sanger sequencing. Primary sequence data were assembled from forward and reverse strands in Geneious version 9.1.8 (https://www.geneious.com). Sequences were aligned in MEGA X using ClustalW [69] and then translated to confirm sequences did not correspond to pseudogenes [70]. The genetic distance (p-distance) within and between species groups was calculated estimating variance with 1000 bootstraps. The p-distance [71] is the proportion of variable nucleotides between two individuals and diverges from the expected genetic distance with time and should be conservative for barcoding.

To define a barcode gap a priori, we sought data from the Barcode of Life Data Systems (BOLD) public repository [72]. Records for Dermaptera *cox1* barcodes were downloaded using the “bold” R package, which downloads complete records for a list of BOLD ID codes. Sequences not identified to species, shorter than 500 bp, and for species represented by fewer than five entries were excluded. Remaining records included *Chelidura guentheri* Galvagni (Forficulidae), *L. riparia*, *N. lividipes*, and both sister species of *F. auricularia*. Sequences were aligned in MEGA X and the genetic distance (p-distance) within and between species was calculated estimating variance with 1000 bootstraps.

### 2.3. Phylogeny

The generic classifications after Steinmann [6] were tested by gathering more sequence data. We obtained a 358 bp fragment of cytochrome *b* (*cob*), a 442 bp fragment of 28S ribosomal DNA (*28S*), and a 466 bp fragment of the tubulin alpha 1 gene (*tuba1*) for selected individuals from the Anisolabididae morphospecies. For each gene fragment, we aimed to obtain at least two individuals for each morphospecies. Cytochrome *b* was amplified and sequenced under the same reaction conditions and cycling profile as *cox1*, using the CB3/CB4 primer pair [73]. The *28S* fragment was amplified with primers designed using Primer3 [74,75] implemented in Geneious using the default settings. Suitable primer binding sites were identified for a fragment > 400 bp with a multiple sequence alignment of four Anisolabididae *28S* sequences from Kamimura [76] in GenBank [77] (accession numbers: AB119545, AB119544, AB119542, AB119549). Two forward (An16F: GAGAAATCCGAATATCTGAA and An17F: AGAAATCCGAATATCTGAAG) and two reverse primers (An481R: AATATAATTGCCAACAATGC and An485R: TGAGAATATAATTGCCAACA) were selected and tested, amplifying a fragment 452–462 bp. PCR was performed using each possible pair, and the combination that produced the strongest bands after staining and electrophoresis was selected for sequencing. *28S* was amplified with a cycle of 93 °C denaturation for three minutes, followed by 30 cycles of 93 °C denaturation for 60 s, 55 °C annealing for 90 s, and 72 °C extension for 75 s, and then 72 °C for four minutes for the final extension. For *tuba1*, the primer pair DDVTubAF/DDVTubAR [78] was used with a cycle of 94 °C denaturation for three minutes, followed by 35 cycles of 94 °C denaturation for 30 s, 50 °C annealing for 30 s, and 72 °C extension for 50 s, followed by 72 °C for six minutes for the final extension. Nuclear genes were amplified, visualised, and sequenced using the same reagent mixtures as mitochondrial ones. Primary sequence data were assembled from forward and reverse strands in Geneious as above. For both *28S* and *tuba1*, ambiguous reads at the ends of the assembly could not be resolved and ends were dropped starting from the first ambiguous read. Sequences were aligned in MEGA X as above, and *cob* and *tuba1* sequences were translated to assure that sequences corresponded to an amino acid sequence without stop codons. In addition to their phylogenetic use, *cob* sequences were also used analogously to *cox1* sequences, estimating the within- and between-species genetic distance for Anisolabididae morphospecies based on the fragment.

Gene trees for each fragment were estimated using the maximum likelihood method in RAxML version 8.2.10 [79] with the standard hill-climbing algorithm, a general time-reversible model of site evolution, and a gamma-distributed rate parameter which was estimated empirically. Mitochondrial fragments (*cox1* and *cob*) were combined into a single mitochondrial data matrix of 1016 bp. Log-likelihoods of estimated trees using rate heterogeneity parameters under gamma and CAT models were compared before selecting the gamma parameter [80]. Node support was estimated with 1000 bootstraps using RAxML’s rapid bootstrapping algorithm. Following this, the sequence alignments were concatenated with SequenceMatrix version 1.8 [81]. A partition homogeneity test (incongruence length difference test) was performed in PAUP* version 4 [82] on the concatenated data to determine whether concatenating the fragments was likely to improve phylogenetic signal. The test failed to reject the null hypothesis of congruence at α = 0.05. The concatenated dataset contained gaps (e.g., *28S* could be obtained for one individual and *tuba1* for another, but both fragments could not be obtained for both individuals). Individuals for which only one fragment was obtained were removed prior to the final analysis. Thomson and Shaffer [83] noted that when data matrices are sparse the identification and removal of “rogue” taxa can improve tree inference and support values. We used the RogueNaRok web service and its eponymous algorithm [84] to identify rogue taxa, pruning potential rogues based on their effect on the majority-rules-consensus tree, but only if pruning that taxon did not drop the number of taxa in a morphospecies below three. Trees were rooted on their outgroups prior to pruning.

Outgroups for all analyses were retrieved from GenBank. *cox1* and *tuba1* sequences for *L. riparia*, *Nala tenuicornis* (de Bormans) (Labiduridae), and *N. lividipes* were originally published in Naegle et al. [18]. The *28S* sequence for *L. riparia* was originally published in Kamimura [76]. *cox1* and *cob* sequences for *Challia fletcheri* Burr (Pygidicranidae) were extracted from a complete sequence of the *C. fletcheri* mitochondrial genome [85]. The *C. fletcheri* mitochondrial genome contains the only publicly available Dermaptera *cob* sequence. While *C. fletcheri* was used as the outgroup for single-gene *cob* ML tree, subsequent analyses identified it as a potential rogue taxon, suppressing the inferential power of the concatenated dataset so it was not used as part of the final outgroup. A final tree was estimated from the concatenated pruned alignment in the same manner as above with rate heterogeneity of site evolution estimated independently for *28S*, *tuba1*, and the mitochondrial data with partitions. A subsequent tree was estimated with a constraint tree imposed corresponding to the two putative genera (*Anisolabis* and *Gonolabis*). The best constrained and unconstrained trees were compared using the Shimodairo-Hasegawa test [86], which compares the log-likelihoods of an input tree against a candidate tree or set thereof.

We complemented the ML analysis with a Bayesian estimate of the species tree. *BEAST [87] is an extension of BEAST that estimates species trees from multi-locus sequence data by modelling the multispecies coalescent of the loci. The analysis estimates gene trees for all loci but embeds them within a shared species tree. While multiple individuals are required per known taxon to estimate effective population size by coalescent modelling, the individuals sequenced for each locus need not be the same. This is advantageous in the present case as some individuals yielded usable sequence data for some loci but not others. We used *BEAST2 executed within BEAST v2.5 [88] to estimate the species tree of the seven Anisolabididae morphospecies.

As only topology was of interest, we implemented a strict molecular clock. Substitution models were estimated using bModelTest version 1.1.2 [89], only considering models that distinguish transitions from transversions to reduce computational load. bModelTest allows the estimation of substitution models simultaneous to tree estimation, and performs model averaging automatically. For the multi species coalescent, we used a linear model with a constant root as the population size function. *BEAST only allows population size to take a gamma prior but allows the estimation of the scale parameter of this gamma prior as a hyperprior [90], which we did. Ploidies were set to diploid autosomal nuclear for *28S* and *tuba1* and mitochondrial for the combined *cox1* and *cob* sequence. We used a conditioned reconstructed process (a birth-death process) as the species tree prior [91], with birth and death rates taking conservative exponential priors with means of two and one, respectively. Other priors, such as those relating to the molecular clock rate and the prior probabilities of substitution models considered by bModelTest, were left at their default. We used BEAUTi version 2.5.1 [92] to generate the .xml file for BEAST. The analysis was run on two independent chains of 5 × 10^7^ generations, sampling every 10^4^ generations. Log and tree files were combined using LogCombiner version 2.5.1 [88], discarding 10% of the states as burn-in. We used Tracer version 1.7.1 [93] to inspect Markov chain traces and determine whether posterior distributions had been adequately explored. The estimated species tree and gene trees were sampled from combined *BEAST output files using TreeAnnotator version 2.5.1 [88] to generate maximum clade credibility trees.

*BEAST requires sequences to be assigned to taxa prior to analysis, and so gene tree priors assign all individuals of the same taxa to a monophyletic group. To compare the topologies of the gene trees estimated with *BEAST to those estimated using RAxML, the RAxML analyses were re-run with the search space constrained to include only trees where the known morphospecies are monophyletic. This is a crude approximation of the process of assigning a prior on tree topology. The unconstrained and constrained RAxML and *BEAST gene trees were plotted in parallel for comparison.

All sequence data were submitted to GenBank under accession numbers MK399433 to MK399688.

All visualisation and data preparation not explicitly mentioned was performed in R 3.5.0 [94] using RStudio version 1.1.453 [95]. We used the *ggtree* R package [96] for all tree figures.

## 3. Results

### 3.1. Sampling, Identification, and Morphology

A total of 136 unknown males were resolved into seven Anisolabididae morphospecies based on combinations of the shapes of the male forceps and parameres (Figure 1a–n). Anisolabididae were found only in SA and WA. The morphospecies could be sorted to two genera, *Anisolabis* Fieber and *Gonolabis* Burr, based on the shape of the parameres, which were either much longer than wide or truncate respectively (Figure 1). Individually, forceps shape was similar for some species (cf. *G. forcipata*, *G*. sp. 2, and *G*. sp. 3; Figure 1e,f,k–n) but the combinations of characters reliably distinguished morphospecies. Most morphospecies could not be identified to species using Steinmann [6], except *G. forcipata* and *G*. nr. *gilesi*. Some morphospecies are easily recognisable by forceps alone. *Anisolabis* sp. 1 is the only morphospecies with a distinct lateral tooth on the forceps (Figure 1a), and *G*. nr. *gilesi* the only one with a dorsoventral tooth (Figure 1g). *Anisolabis* sp. 2 and *G*. sp. 1 have broadly similar forceps shapes (Figure 1c,i), with the former distinguished by the parallel region towards the base of the forceps. Summary statistics for morphometric variables (derived variables and original measurements) are provided in the Appendix A.

Principal component analysis (PCA) reduced variation across the four morphometric variables to three principal components (PCs) that cumulatively explained 94.5% of the total variance (41.7%, 27.6% and 25.2% respectively) (Table 1) among individuals. The first PC is dominated by the forceps asymmetry and the paramere ratio although the other two variables show relatively high loadings, and the second PC by the basal width of the forceps and the paramere ratio (Table 1). A biplot of the first two PCs revealed that the two putative *Anisolabis* spp. are easily separable from the putative *Gonolabis* spp. by morphometrics, although *G*. nr. *gilesi* shows some overlap with *Anisolabis* sp. 1 (but is otherwise distinguishable). *Anisolabis* sp. 2 is clearly distinguished from all other morphospecies. The *Gonolabis* spp. show much overlap. *Gonolabis* sp. 3, of which only 3 males were present in the sample, showed greater variation between individuals than other *Gonolabis* morphospecies, and is not easily distinguishable by these morphometrics, showing overlap with *G*. sp. 2 and *G. forcipata* (Figure 2).

Discriminant function analysis reliably distinguished the two putative genera and the two *Anisolabis* species with 100% accuracy, but accuracy was only 71.28% (95% CI: 61.02–80.14%) for the *Gonolabis* spp., which increased to 75.82% (65.72–84.19%) when *G*. sp. 3 was excluded. *Gonolabis* nr. *gilesi* was classified with the least accuracy, in 57.14% of cases. The analysis classified *G. forcipata* with 73.913% accuracy and *G*. sp. 1 with 90.625% accuracy, invariant to the inclusion of *G*. sp. 3, while the classification of *G*. sp. 2 improved from 59.091 to 68.182% when *G*. sp. 3 was excluded. *Gonolabis* sp. 3 was not correctly classified in any case when it was included. The most informative variables of the discriminant functions varied by the classification being performed. To distinguish genus, the forceps asymmetry and paramere ratio were the most informative; for distinguishing the *Anisolabis*, the forceps ratio was the most informative; and for the *Gonolabis* classification, the most informative variable was the paramere ratio (Appendix A). However, the data did not satisfy multivariate normality (Royston’s test of multivariate normality; H = 147.286, *p* < 0.001), even when split by genus (*Anisolabis*, H = 14.005, *p* = 0.001; *Gonolabis*, H = 62.871, *p* < 0.001).

Barcode sequences indicated that five of the seven male morphospecies had female representatives in the sample, and these were used to define female morphology for *A*. sp. 1, *A*. sp. 2, *G. forcipata*, *G*. nr. *gilesi*, and *G*. sp. 1. Female *A*. sp. 1 are easily recognisable because of their broad morphological similarity to the males; they are similarly patterned and coloured, and their forceps also bear the same median lateral tooth (cf. Figure 1a), although, as is the case for all females, their forceps were otherwise straight and not rounded like their male counterparts. Of the two WA morphospecies, female *G*. sp. 1 may be distinguished from those of *A*. sp. 2 by the darker sternites, and the colour pattern of the femur. *Anisolabis* sp. 2 femurs are proximally darker (towards the trochanter) and distally lighter (towards the tibia), and *G*. sp. 1 femurs are either the inverse or uniformly coloured. The ultimate sternites of the two are both setose, although the setae of *G*. sp. 1 are much longer and more prominent. Of the SA morphospecies, *G*. nr. *gilesi* may be distinguished from *G. forcipata* by a small dorsoventral tooth in the same position as that of the males (c.f. Figure 1g). The remaining females were counted, making the final count of adult Anisolabididae 206. A key to distinguish both males and females of the morphospecies examined in this study is provided in Appendix B. Most morphospecies were found at relatively restricted times of the year. *Anisolabis* sp. 2, *G*. *forcipata*, *G*. sp. 1, and *G*. sp. 2, and *G*. sp. 3 were all only found in winter. *Anisolabis* sp. 1 and *G*. nr. *gilesi* were found throughout the year. Some morphospecies were more common in particular years. Ninety percent of *G*. *forcipata* specimens were taken from 2016 collections. All but one of the *G*. sp. 2 specimens were found in 2016, as were all of the *G*. sp. 3 specimens. Additionally, some morphospecies were found only at a restricted set of sites. *Anisolabis* sp. 2, *G*. *forcipata*, *G*. nr. *gilesi*, and *G*. sp. 1 were found primarily at the more regularly sampled sites (#2, #3, #19, #20) (Appendix A and Appendix A). In these cases, the morphospecies were only found at the time of year when the full set of sites was sampled, so their abundance would not have been inflated by year-round sampling. The exception is *A*. sp. 1, which appeared in the sampling throughout the year.

Morphospecies occurrences were mapped (Figure 3). The Anisolabididae fauna of WA and SA showed little overlap. *Anisolabis* sp. 2 and *G*. sp. 1 were only found in WA, and *G. forcipata*, *G*. nr. *gilesi*, *G*. sp. 2 and *G*. sp. 3 only in SA. *Anisolabis* sp. 1 was the only species common to both (Figure 3). In SA, *G. forcipata* and *G*. nr. *gilesi* were only found east of Adelaide; *G. forcipata* was found north of Lake Alexandria and *G*. nr. *gilesi* to the south, and together at only one site (Figure 3). The remaining SA species were only found to the west of Adelaide (Figure 3). *Gonolabis* sp. 2 and *G*. sp. 3, which show morphological similarity (Figure 1), were found together at two localities (Figure 3). In WA, *G*. sp. 1 and *A*. sp. 1 were the sole morphospecies at two sites respectively, and *G*. sp. 1 and *A*. sp. 2 were the most abundant at the remaining two sites (Figure 3).

### 3.2. Barcoding

*Forficula auricularia* was the most common species found in the BOLD public repository, the sister species (A and B) totalling 688 records. The next most abundant species (*C. guentheri* and *L. riparia*) totalled six each. *Forficula auricularia* A had more than four times as many records as B (Appendix A). Intraspecific comparisons all showed <0.01 mean p-distance with similarly low variance and all interspecific comparisons showed >0.1 mean p-distance (Appendix A). The lowest pairwise group mean p-distances were 0.102 between the two *F. auricularia* species, and 0.147 between *L. riparia* and *N. lividipes*, which are in the same family.

The number of sequences obtained from field samples varied, although the full 658 bp barcode fragment was acquired for all included individuals (Table 2). Intraspecific genetic distance was <0.01 for most morphospecies, and was 0.017, 0.040, and 0.013 for *G*. *forcipata*, *G*. sp. 2, and *G*. sp. 3, respectively (Table 2). Interspecific genetic distances were all >0.1, with the lowest values found among the Anisolabididae (Table 2). The barcode sequences obtained corroborated the morphospecies defined after Steinmann (1989b); all within and between-species comparisons were consistent with a barcode gap. Genetic distances at the *cob* locus were similar; intraspecific genetic distances were all <0.020 except for *G*. sp. 1 which showed 0.054 (Table 3), although this was well outside the range of interspecific distances which were all greater than 0.180, except for the pairwise comparison of *G*. nr. *gilesi* and *G*. sp. *1*, which had a p-distance of 0.106 between them (Table 3). A BLASTn search of all *L. truncata* barcode sequences showed pairwise similarities of 77.26–83.09% to *L. riparia cox1* sequences (accession numbers: AB435163, AY555549, JN241998).

### 3.3. Phylogeny

The number of sequences obtained varied by morphospecies and gene; at least two morphospecies were covered for each gene except for *tuba1*. Only one *tuba1* sequence was obtained from *G*. nr. *gilesi* and *G*. sp. 2. The final concatenated dataset, pruned of potential rogue taxa, comprised 39 individuals across the seven morphospecies and four outgroup taxa in the Labiduridae and Pygidicranidae families (Table 4). The concatenated ML gene tree resolved all Anisolabididae morphospecies to clades with 100% bootstrap support, and so the tree is presented with morphospecies-internal branches reduced for clarity, opposite the *BEAST species tree (Figure 4). The one exception is *G*. sp. 3 which, although reliably split from *G*. sp. 2, has low bootstrap support for the node splitting the two individuals. Branch lengths are not shown to emphasise topology. Outgroups have been pruned in the final figures. The ML and Bayesian estimates are mostly identical; *A*. sp. 1 is sister to all other taxa, followed by *G. nr. gilesi* (Figure 4). In both trees, *G. forcipata*, *G*. sp. 2, and *G*. sp. 3 form a clade with high support (Figure 4) (referred to henceforth as the SA clade). The *BEAST tree resolves *G*. sp. 1 and *A*. sp. 2 to a highly supported clade, sister to the SA clade (Figure 4). The ML tree places *G*. sp. 1 and sister to the SA clade and *A*. sp. 2, and *A*. sp. 2 as sister to the SA clade, but with low bootstrap support (Figure 4). While not annotated on the tree, the branches preceding the nodes with the lowest bootstrap support are also some of the shortest in the tree, both at 0.028.

Regardless, the generic assignments following Steinmann [6] are paraphyletic and polyphyletic for *Gonolabis* and *Anisolabis* respectively according to the molecular phylogeny. The log-likelihoods of the best-known ML trees unconstrained and constrained by the monophylies of *Gonolabis* and *Anisolabis* were −8205.895 and −8216.307, respectively. The Shimodairo-Hasegawa test did not indicate that the constrained tree was significantly worse (*p* > 0.05), although plotting it did reveal that the constraint caused many of the branches to reduce to almost zero. This would raise the constrained tree’s log-likelihood considerably [71], so the result of this test may be unreliable. *Gonolabis forcipata*, *G*. sp. 2 and *G*. sp. 3 form a highly supported clade (Figure 4) and they also show considerable morphometric overlap (Figure 2). All other individual morphospecies form their own clades. Each of the clades proposed by *BEAST is distinguished by distinct forceps morphology (c.f. Figure 1 and Figure 4). Moreover, there appear to be two state-specific clades, the SA clade and that formed by *G*. sp. 1 and *A*. sp. 2 (c.f. Figure 3 and Figure 4) (referred to henceforth as the WA clade).

Gene trees were less straightforward, although those estimated by *BEAST showed generally higher node support than the ML trees, both unconstrained and constrained by the monophyly of the morphospecies. Across all gene trees, *A*. sp. 1 was sister to all other taxa (Appendix A). The SA and WA clades are also mostly present, although the former across more of the gene trees but it should be noted that the nuclear ML gene trees do not consistently indicate divergence of *G*. sp. 2 and *G*. sp. 3 (Appendix A). *BEAST estimated highly supported topologies identical to the species tree for the *tuba1* and the combined mitochondrial datasets (c.f. Figure 4, Appendix A. The *BEAST *28S* tree was similar to the concatenated ML tree, with the places of *G*. sp. 1 and *A*. sp. 2 swapped (c.f. Figure 4 and Appendix A).

## 4. Discussion

This study assessed the current state of taxonomic knowledge available to enumerate the diversity of dermapteran fauna in the southern Australian grain growing regions and is the first to apply molecular methods to study Australian native Dermaptera. Anisolabididae were classified based on combinations of male primary and secondary sexual characters. These reliably sorted males into morphospecies, all of which showed consistently lower genetic distance within than among putative taxa at the *cox1* barcode locus as well as a fragment of *cob*, reflecting a clear barcode gap. The use of barcodes therefore shows promise for the study of Australian Dermaptera. Importantly, this allowed females of the morphospecies to be identified, which permitted a more complete assessment of morphology and allowed all samples to be resolved. Female genitalia, the spermatheca and the ovipositor, have been recognised as potentially important taxonomic characters [97,98], and so future studies should incorporate these characters into their morphological assessment, as they will likely provide more clarity to the problem of female identification. Another important caveat is the lack of female specimens for *G*. sp. 2 and *G*. sp. 3. Given the morphological similarity within the SA (*G. forcipata*, *G*. sp. 2, *G*. sp. 3) clade, females of *G. forcipata* could be mistaken for females of these two unknown species, over-inflating estimates of the distribution and abundance of *G. forcipata*. This is possible, but unlikely, as *G. forcipata* was not found at any of the sites where male *G*. sp. 2 and *G*. sp. 3 were collected. Moreover, juvenile morphology remains opaque, although cost-effective adult-larval matching solutions for large numbers of samples are available [99], which could take advantage of the sequence data reported herein. Given that these morphospecies were found at the sites with less-frequent sampling, it is likely that the scarcity of these morphospecies in the sample is an effect of sampling bias.

### 4.1. The Utility of Barcoding for Earwigs

Barcoding in the strict sense (matching barcode sequences of unknown specimens to those of known specimens) is of little use at present for Dermaptera in Australia and worldwide. Two species (*F. auricularia* B and *N. lividipes*) in the sample explored in this study were represented on the BOLD public repository and these were both readily identifiable by morphology in any case. In the absence of a previously sampled conspecific, an unknown specimen may be assigned correctly to genus, tribe, or family if the higher taxon is sufficiently sampled [100] but public BOLD records include only 16 individuals from the Anisolabididae family, the majority being either *A. maritima* or unidentified specimens from the *Anisolabis* and *Euborellia* Burr genera. GenBank is equally depauperate, although it does contain 24 *cox1* sequences for *Anisolabis littorea* (Whiting) from Goldberg and Trewick [101]. A BOLD identification search of a 658 bp *L. truncata* barcode sequence returned no hits, despite six known congeneric records (*L. riparia*). This echoes the findings of Kwong et al. [102] and Kvist [103]; barcode reference databases show taxonomic skew at several scales, uneven coverage, and many unidentified specimens for which no additional data (e.g., photos, voucher information) are provided. A GenBank BLASTn search returned *L. riparia* and *Labidura japonica* (Haan) as close matches, although members of other orders showed equal or higher sequence similarity as well. The sequence reported herein appear to be the first *L. truncata* barcode sequences made publicly available. Pairwise distances to GenBank accessions support the distinction of *L. truncata* as a separate species to *L. riparia* [34].

Most of the species-identified Dermaptera records available via BOLD list either in-house identification methods (i.e., BOLD’s Barcode Index Number system or tree-based methods) and the majority that report morphological identification use only digitised photos submitted to BOLD, not type material. The taxonomic literature consulted for species diagnoses in BOLD records is also not given. For the current set of earwig records, this is not necessarily problematic; the majority of records represent the well-sampled *F. auricularia* species complex. However, for uncharacterised species such as those considered here, it is important to report how a specimen was identified, provide citations of the authoritative texts and keys and, where possible, original descriptions [104]. This is especially relevant for taxa with significant taxonomic impediment such as the Dermaptera where the currently accepted Latin binomial and familial placement may require considerable command of a literature that is difficult to access, sparse, and occasionally self-contradictory (see [9]).

Morphological assessments were equally challenging. Most specimens could not be identified to a known species using Steinmann [6], suggesting that they are undescribed, or the available information is of insufficient detail. *Gonolabis forcipata* was the only species that could be confidently identified, but type information only lists south-western WA whereas the specimens in the present study are from SA. All type specimens for described species are in Europe (see [9]), so it is difficult to determine whether SA represents a new location for this species.

### 4.2. Distribution of Anisolabididae

The complete distribution of the Anisolabididae remains unclear at present, although some inferences can be made. Aside from *A*. sp. 1, no morphospecies were shared between WA and SA, and SA shows a distinct fauna east and west of Adelaide. These patterns need to be confirmed with more sampling, particularly given that this study’s scope was limited (by design) to grain growing regions. Extended sampling is needed to better capture the landscape diversity of southern WA and SA, especially along the coast of the Great Australian Bight. Given that agricultural transformations have negative impacts on species richness in general [105,106] and that the intensity of cultivation has a negative effect on arthropod species richness specifically [107,108,109], extending sampling to a broader range of habitats outside agricultural landscapes is likely to further increase the known diversity of Australian Anisolabididae. The Bayesian species tree estimate divides five of the seven Anisolabididae morphospecies into two state-restricted clades, a WA clade and an SA clade. Australian insect fauna often shows such short-range endemism, particularly in taxa with poor dispersal ability [110]. Anisolabididae are likely to be poor dispersers (as are most of the order Dermaptera) because of their small size, litter-dwelling habit, and wingless habitus, although there are the notable cosmopolitan exceptions *A. maritima* and *Euborellia annulipes* (Lucas). It is curious that *E. annulipes* was absent from the sample. It was most recently reported as being a greenhouse pest [23]; highly seasonal, broadacre crop environments may not provide suitable habitat for this species or perhaps it will appear with additional sampling.

The seasonal distribution of the morphospecies is relevant to their potential for pest control. Most morphospecies were found as adults only in the winter, when grain crops in southern Australia are grown [111]; in other seasons, assemblages of earwigs that act as natural enemies may recede into surrounding vegetation. Enumeration of the roles of the morphospecies within the agroecosystem and their fine-scale distribution in the landscape may contribute to integrated control of pests of Australian grains [112].

### 4.3. Conflict between Morphological and Molecular Inferences

Neither maximum-likelihood nor Bayesian multilocus phylogenetic approaches supported the two putative genera based on paramere morphology. However, there appear to be state specific clades distinguished by forceps morphology. All clades have unique male forceps morphology, highlighting the trait’s utility in distinguishing morphospecies, but in some cases paramere morphology is necessary to distinguish species. At broader scales, across clades, sole reliance on paramere morphology might lead to misleading species assignment. Some of the *G*. sp. 1 and *G*. sp. 2 specimens showed similar paramere morphology despite being phylogenetically distant. Moreover, the lengths of the parameres of *A*. sp. 1 and *A*. sp. 2 are closer to each other than to other morphospecies, but the two species are not closely related according to the molecular phylogeny. This echoes the argument made by Kamimura et al. [113] working with another family, the Pygidicranidae; even among closely related congeners, genital characters may mislead.

The generic assignments here are not all necessarily valid. Srivastava [60] revised the systematics of the Anisolabididae genera and generic assignments based on this system would yield the following changes; *A* sp. 1 would become *Gonolabis* sp.; *Anisolabis* sp. 2 would become *Apolabis* sp.; *G. forcipata* remains unknown, keying to either *Mongolabis* or *Anisolabella*, but lacking the snout-like paramere structure of *Mongolabis* nor the continuous outer paramere margin of *Anisolabella*; *G* nr. *gilesi* would become *Anisolabella* sp. 1; *G*. sp. 1 would become *Mongolabis* sp.; *G*. sp. 2 would remain unknown for similar reasons to *G. forcipata*; and *G*. sp. 3 would become *Anisolabella* sp. 2. In such a case, the *Anisolabella* become polyphyletic, and the conjecture that similar paramere structure does not reflect phylogeny remains, regardless of the nomenclature used. This is unsurprising given Srivastava’s heavy reliance on paramere morphology for distinguishing the genera of the Anisolabidinae subfamily [60]. Given these issues, the specimens presented here should be incorporated, along with their putative identifications, into a fuller cladistic and phylogenetic analysis.

The phylogenetic signal of insect genitalia has been subject to some debate. Polyandry has been found in confamilial species *Euborellia plebeja* (Dohrn) [114,115,116,117] and *Euborellia brunneri* (Dohrn) [118], so it is likely that genital morphology is subject to sexual selection. However, the precise function of the parameres is unknown [41] so it is difficult to make definitive statements about the effect of sexual selection on their morphology. Structures of the male forceps often play specific roles in male-male competition [38] so they should also be under strong sexual selection for morphological evolution. Eberhard [119] and Arnqvist and Rowe [120] both cited Losos [121] in arguing that rates of morphological evolution (which may be accelerated by sexual selection) in insect male genitalia relative to the rate of speciation should be high enough to swamp phylogenetic inertia and that there should therefore be little correlation between morphological similarity in male genitalia and phylogenetic distance. The present study is equivocal on the matter, as the SA clade species show similar paramere morphology, while the WA clade do not. Song and Bucheli [122] explicitly tested the phylogenetic signal of male insect genitalia and found that the genitalia are composite structures, and some sub-components may be conserved whereas others may be free to diversify. In the present study, the parameres were used for generic identification, but other characters such as the length and asymmetry of the virgae and the length of the penis lobes may also prove useful in describing the phylogeny of the morphospecies as they have already in describing the family-level phylogeny of the order [41]. A full cladistic analysis of the composite structure of Dermaptera genitalia may shed new light on the heretofore rather unstable taxonomy.

## 5. Conclusions

This study presents, to our knowledge, the most detailed study of Australian Anisolabididae to date. We combined extant taxonomic knowledge, the fidelity of which is unknown, with linear morphometrics, simple molecular analyses of DNA barcodes, and more detailed phylogenetic analyses. Our findings demonstrate that the current state of knowledge of Australian Anisolabididae is sufficient to distinguish male morphospecies from each other, but accurate species and generic assignment is problematic. We provide a foundation for future dermapteran biodiversity research and taxonomic revision of a morphologically challenging family. A future study that provides comprehensive geographic and habitat coverage of the sampled regions, particularly the south-eastern coast of WA, the regions surrounding the Spencer Gulf in SA, and the coast of the Great Australian Bight, would likely uncover considerable dermapteran diversity. The current research also raises the possibility of region-specific species of beneficial earwigs providing fortuitous pest control in grain systems and highlights the importance of detailed studies to fully enumerate local insect biodiversity.

## Figures and Tables

**Figure 1 insects-10-00072-f001:**
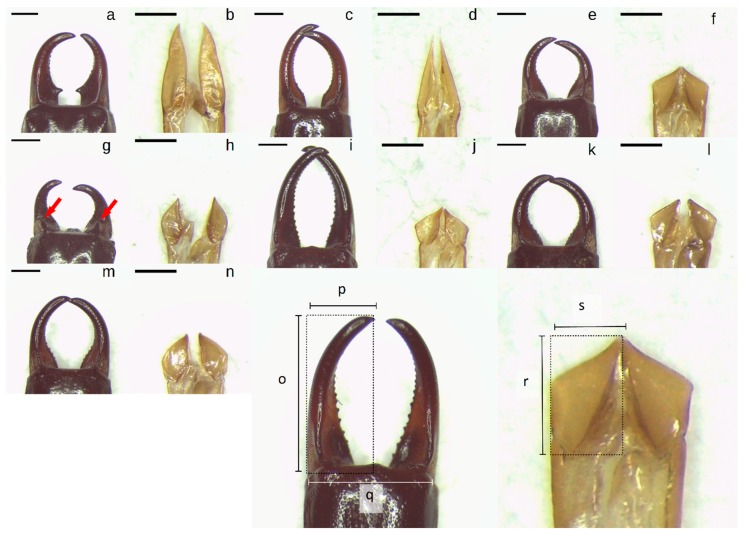
(**a**–**n**): Photos of male Anisolabididae forceps and parameres. (**a**,**b**): *Anisolabis* sp. 1. (**c**,**d**): *Anisolabis* sp. 2. (**e**,**f**): *Gonolabis forcipata* Burr. (**g**,**h**): *Gonolabis* nr. *gilesi* Steinmann. (**i**,**j**): *Gonolabis* sp. 1. (**k**,**l**): *Gonolabis* sp. 2. (**m**,**n**): *Gonolabis* sp. 3. (**o**–**s**): Diagrams of morphometric measurements taken. (**o**): Forceps length. (**p**): forceps width. (**q**): basal width of forceps. (**r**): paramere length. (**s**): paramere width. All scale bars indicate 1 mm. Red arrows indicate location of dorsoventrally oriented teeth on forceps of male *G*. nr. *gilesi*.

**Figure 2 insects-10-00072-f002:**
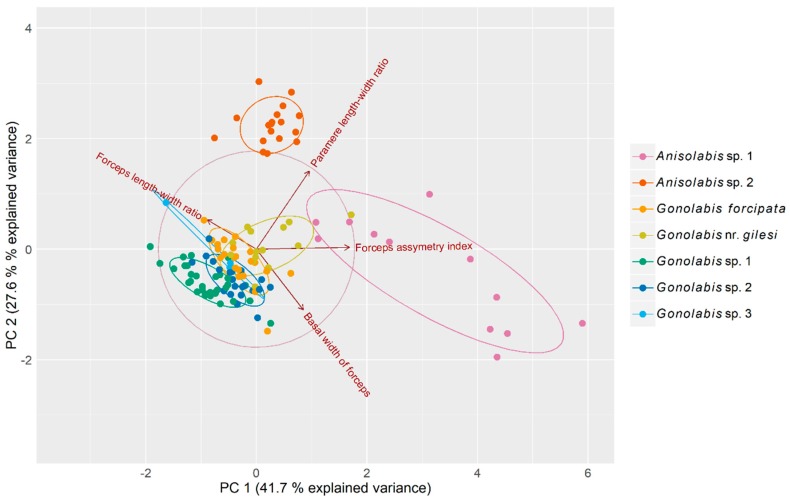
Biplot of principal component analysis measuring variation among male Anisolabididae specimens for seven morphometric variables related to primary and secondary male sex characteristics. Ellipses represent 68% normally distributed confidence intervals for the principal components plotted to aid visualisation of clusters. Vectors with arrows represent loadings of the variables on the principal components. Their direction relative to the x- and y-axes and each other represents their correlations with each principal component and each other. Orthogonal vectors are not correlated and vectors in opposing directions are negatively correlated. The length of the vectors in relation to the correlation circle is a relative indicator of the quality of each variable’s representation in the biplot.

**Figure 3 insects-10-00072-f003:**
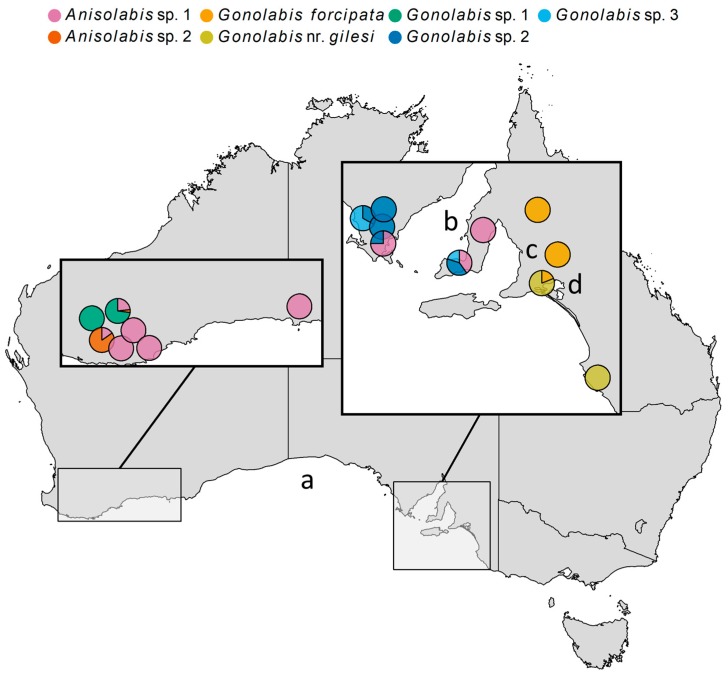
Map of Australia showing locations of collection of Anisolabididae specimens, with pie charts indicating the proportion of each morphospecies found at each site. Some pies have been nudged slightly from their original position to prevent overlapping. Left-inset: Western Australia (WA) sites. Right-inset: South Australia (SA) sites. (**a**) Great Australian Bight. (**b**) Spencer Gulf. (**c**) Adelaide. (**d**) Lake Alexandria.

**Figure 4 insects-10-00072-f004:**
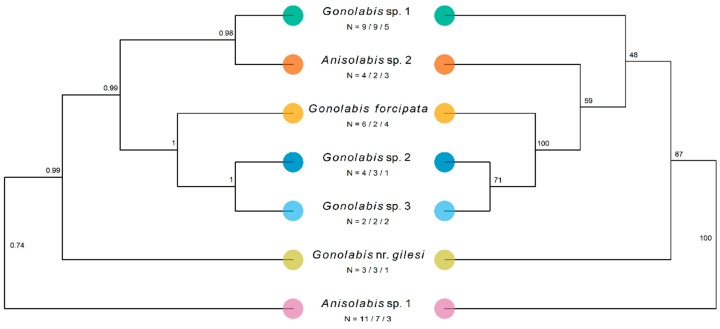
Results of Bayesian and maximum-likelihood phylogenetic analysis estimating the species tree of the Anisolabididae morphospecies. Labels under species names represent the number of individuals of that morphospecies from which sequences of a combined mitochondrial fragment of cytochrome *c* oxidase subunit 1 (658 bp) and cytochrome *b* (358 bp) (1016 bp), a fragment of 28s ribosomal DNA (*28S*) (442 bp), and the tubulin alpha-1 gene (*tuba1*) (468 bp) were obtained. Trees were rooted using members of the Labiduridae family, but these were pruned for visualisation. Left: Maximum clade credibility species tree of Anisolabididae morphospecies estimated using *BEAST, a Bayesian reconstruction of the multi-species coalescent of the individual genes. Node labels indicate posterior probability. 95% HPDI of tree likelihood = [−7949.638, −7421.390], mean tree likelihood = −7446.813. Right: Maximum likelihood tree for a concatenated dataset of fragments of the mitochondrial (*cox1* + *cob*), *28S*, and *tuba1* sequence data inferred using RAxML with a general time-reversible model of sequence evolution with a gamma-distributed rate parameter. Model parameters were estimated separately for each fragment using a partition. Node labels show bootstrap support from 1000 bootstraps using RAxML’s rapid bootstrapping algorithm. Tree likelihood = −8205.895. NB: While *BEAST estimates a species tree with as many leaves as taxa considering each gene tree separately, RAxML reconstructs a tree using the whole concatenated dataset, and infers the topological placement of all individuals. As all morphospecies except *G*. sp. 3 formed clades with 100% bootstrap support, these have been collapsed for comparison with the *BEAST species tree such that the ML tree only shows morphospecies and not individuals.

**Table 1 insects-10-00072-t001:** Summary of principal component analysis measuring variation among male Anisolabididae specimens for seven morphometric variables related to primary and secondary male sex characteristics. Only principal components accounting for more than 10% of the total variation are shown. Component loadings with an absolute value greater than 0.3 are bolded. “Forceps asymmetry” is an index calculated as the sum of the absolute differences between the heights and widths of a specimen’s two forceps, “Paramere length-width ratio” is the ratio of a specimen’s right paramere’s length to its width, “Forceps asymmetry index” refers to an index calculated using the differences between the lengths and width of the two forceps, and “Forceps length-width ratio” is the ratio of the maximum forceps length to the maximum forceps width”. These three measurements were dimensionless while the basal width of the forceps was in mm.

	PC1	PC2	PC3
Component Loadings			
Basal width of forceps	**0.370**	**−0.587**	**0.593**
Forceps asymmetry index	**0.733**	0.017	0.026
Paramere length-width ratio	**0.417**	**0.757**	0.163
Forceps length-width ratio	**−0.389**	0.285	**0.788**
Eigenvalues			
Standard deviation	1.292	1.051	1.003
Variance explained	0.417	0.276	0.252
Cumulative variance explained	0.417	0.693	0.945

**Table 2 insects-10-00072-t002:** Uncorrected p-distances within and between Dermaptera morphospecies cytochrome *c* oxidase subunit 1 barcodes (658 bp). Interspecific means and variances are shown on the left and right of the diagonal. Intraspecific mean shown on diagonal with variance in brackets. Variances were calculated using 1000 bootstraps. N refers to number of individuals of morphospecies sequenced. Comparisons between morphospecies in the Anisolabididae family are delineated by a hashed line. *Forficula auricularia* refers to clade B from Wirth et al. [58], as clade A is not found in Australia [59].

	N	*Forficula auricularia*	*Labidura truncata*	*Nala lividipes*	*Anisolabis* sp. 1	*Anisolabis* sp. 2	*Gonolabis forcipata*	*Gonolabis* nr. *gilesi*	*Gonolabis* sp. 1	*Gonolabis* sp. 2	*Gonolabis* sp. 3
*Forficula auricularia*	39	0.009 (0.003)	0.015	0.016	0.015	0.016	0.016	0.015	0.015	0.015	0.015
*Labidura truncata*	33	0.222	0.007 (0.002)	0.014	0.014	0.015	0.015	0.015	0.014	0.014	0.015
*Nala lividipes*	26	0.227	0.181	0.005 (0.001)	0.014	0.015	0.015	0.015	0.014	0.015	0.016
*Anisolabis* sp. 1	17	0.220	0.178	0.194	0.008 (0.002)	0.012	0.013	0.013	0.012	0.013	0.013
*Anisolabis* sp. 2	5	0.228	0.200	0.200	0.105	0.001 (0.001)	0.013	0.013	0.012	0.013	0.015
*Gonolabis forcipata*	8	0.226	0.209	0.199	0.145	0.154	0.017 (0.003)	0.014	0.013	0.014	0.015
*Gonolabis* nr. *gilesi*	3	0.222	0.198	0.202	0.136	0.139	0.179	0.002 (0.001)	0.011	0.013	0.014
*Gonolabis* sp. 1	12	0.216	0.189	0.196	0.117	0.121	0.151	0.104	0.040 (0.007)	0.011	0.014
*Gonolabis* sp. 2	4	0.217	0.196	0.214	0.151	0.154	0.178	0.151	0.115	0.013 (0.002)	0.013
*Gonolabis* sp. 3	2	0.238	0.199	0.221	0.139	0.162	0.182	0.195	0.169	0.166	0.000 (0.000)

**Table 3 insects-10-00072-t003:** Uncorrected p-distances within and between Anisolabididae morphospecies cytochrome *b* sequences (358 bp). Interspecific means and variances are shown on the left and right of the diagonal. Intraspecific mean shown on diagonal with variance in brackets. Variances were calculated using 1000 bootstraps. N refers to number of individuals of morphospecies sequenced.

	N	*Anisolabis* sp. 1	*Anisolabis* sp. 2	*Gonolabis forcipata*	*Gonolabis* nr. *gilesi*	*Gonolabis* sp. 1	*Gonolabis* sp. 2	*Gonolabis* sp. 3
*Anisolabis* sp. 1	11	0.016 (0.004)	0.019	0.022	0.021	0.020	0.023	0.022
*Anisolabis* sp. 2	5	0.186	0.001 (0.001)	0.022	0.020	0.020	0.022	0.023
*Gonolabis forcipata*	8	0.249	0.267	0.020 (0.005)	0.022	0.022	0.023	0.024
*Gonolabis* nr. *gilesi*	3	0.230	0.214	0.254	0.001 (0.001)	0.014	0.021	0.022
*Gonolabis* sp. 1	16	0.221	0.194	0.268	0.106	0.054 (0.012)	0.020	0.021
*Gonolabis* sp. 2	5	0.282	0.258	0.305	0.221	0.217	0.019 (0.004)	0.023
*Gonolabis* sp. 3	2	0.234	0.262	0.315	0.265	0.252	0.294	0.004 (0.003)

**Table 4 insects-10-00072-t004:** Accession numbers of sequences generated by this study and those retrieved from GenBank for phylogenetic analysis. Dashes indicate missing data. ‘*cox1*’ refers to a fragment of cytochrome *c* oxidase subunit 1 (658 bp). ‘*cob*’ refers to a fragment of cytochrome *b* (358 bp). ‘*28S*’ refers to a fragment *28S* ribosomal DNA (442 bp). ‘*tuba1*’ refers to a fragment of the tubulin alpha-1 gene (466 bp). ‘Individual’ refers to a single adult Anisolabididae specimen’s in-house code number and is provided for reference to Appendix A.

Individual	Morphospecies	Family	*cox1*	*cob*	*28S*	*tuba1*
167	*Anisolabis* sp. 1	Anisolabididae	MK399508	MK399441	MK399637	-
173		Anisolabididae	MK399511	MK399442	MK399638	-
253		Anisolabididae	MK399540	MK399447	MK399643	-
254		Anisolabididae	MK399541	MK399448	MK399644	MK399671
269		Anisolabididae	MK399545	MK399449	-	MK399672
298		Anisolabididae	MK399554	-	-	MK399675
301		Anisolabididae	MK399558	MK399453	-	-
306		Anisolabididae	MK399562	MK399454	-	-
344		Anisolabididae	MK399584	MK399464	MK399652	-
347		Anisolabididae	MK399587	MK399465	MK399653	-
348		Anisolabididae	MK399588	MK399466	MK399654	-
153	*Anisolabis* sp. 2	Anisolabididae	MK399502	MK399435	-	MK399665
155		Anisolabididae	MK399504	MK399437	-	MK399667
160		Anisolabididae	MK399505	MK399438	MK399635	-
161		Anisolabididae	MK399506	MK399439	MK399636	MK399668
288	*Gonolabis forcipata*	Anisolabididae	MK399548	MK399452	-	MK399674
324		Anisolabididae	MK399574	MK399458	-	MK399678
326		Anisolabididae	MK399575	MK399459	MK399649	MK399679
327		Anisolabididae	MK399576	MK399460	MK399650	-
331		Anisolabididae	MK399578	MK399462	-	-
333		Anisolabididae	MK399579	MK399463	-	MK399681
308	*Gonolabis* nr. *gilesi*	Anisolabididae	MK399564	MK399456	MK399647	MK399677
309		Anisolabididae	MK399565	MK399457	MK399648	-
484		Anisolabididae	MK399610	MK399471	MK399658	-
146	*Gonolabis* sp. 1	Anisolabididae	MK399500	MK399433	MK399632	-
152		Anisolabididae	MK399501	MK399434	MK399633	-
384		Anisolabididae	MK399598	MK399467	MK399655	MK399682
385		Anisolabididae	MK399599	MK399468	MK399656	MK399683
386		Anisolabididae	MK399600	MK399469	MK399657	-
533		Anisolabididae	MK399613	MK399473	MK399659	MK399684
535		Anisolabididae	MK399614	MK399475	MK399660	-
536		Anisolabididae	MK399615	MK399476	MK399661	MK399685
539		Anisolabididae	MK399618	MK399479	MK399664	MK399688
232	*Gonolabis* sp. 2	Anisolabididae	MK399531	MK399443	MK399639	-
236		Anisolabididae	MK399532	MK399444	MK399640	MK399669
239		Anisolabididae	MK399533	MK399445	MK399641	-
274		Anisolabididae	MK399546	MK399450	-	-
251	*Gonolabis* sp. 3	Anisolabididae	MK399539	MK399446	MK399642	MK399670
276		Anisolabididae	MK399547	MK399451	MK399645	MK399673
C_fletch	*Challia fletcheri*	Pygidicranidae	NC_018538		-	-
L_ripar	*Labidura riparia*	Labiduridae	KX069089	-	AB119553	KX069030
N_tenui	*Nala tenuicornis*	Labiduridae	KX069090	-	-	KX069055
N_livid	*Nala lividipes*	Labiduridae	KX069069	-	-	KX069048

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
