# Peer review of "Morphological and Molecular Analysis of Australian Earwigs (Dermaptera) Points to Unique Species and Regional Endemism in the Anisolabididae Family"

_insects, 2019, doi:10.3390/insects10030072_

Round 1
Reviewer 1 Report
The authors present results of interesting and valuable research focused on the diversity of southern Australian Anisolabididae in agriculture ecosystem. The authors distinguished 7 morphospecies by morphometric analyses and than applied cox1 barcodes (and next DNA fragments) to corroborate identifications and assess the monophyly of studied genera.
The paper is well written, the style is precise and the results are well supported by statistical analyses. However, I have some questions and suggestions for authors - see the comments below.
Comment 1: The Anisolabididae have not been yet assessed using numerical or cladistic methods. The taxonomic positions of individual genera are uncertain, as well as the phylogenetic relationships within the family. Therefore the taxonomy of Anisolabididae needs to be consider as provisional and a rigorous phylogenetic analysis of the subfamily based on molecular evidence is necessary to clarifying the relevance and the relationships of the genera that have been described to date. Diagnoses of individual genera are based on length and shape of male parameres, but it is evident, that such system si rather artificial. Anyway, there are more classifications than that published by Steinmann (1989), and evaluated MS dealing with the intergenera relationships needs to contain more precise discussion of this problematics. At least, the most recent classification of Srivastava (1999), used also in his later monograph (Steinmann 2003), should be mentioned and discussed, because according to this classification, based exclusively on male genitals, all species of Australian Gonolabis mentioned in evaluated paper belong to genus Mongolabis (!). Another source, Sakai (2000), is worth mentioning.
Comment 2: Earwigs (incl. Anisolabididae) showing high variability in body length, therefore the linear morphometric measures used by the authors are probably correlated with the body length of particular individual (possibly rather than with species). Especially, length and width of the parameres, and maximum length and width of the forceps are cases characters associated with the body robustness. And the body robustness could be associated with local (actual) environmental conditions, esp. unfavourable climatic fluctuations, as low humidity or high temperature. Such conditions could lead to higher ratio of small specimens in the population (not only within, but also across syntopically occurring species). Besides that, forceps are differently curved in males of some Anisolabididae species in relation to their size – small specimens show nearly parallel-sided forceps, bigger specimens hooked.
I recommend to test the dependency of measured characters on the body size in studied species. If the lengths are dependent on body size, I suggest replacing all these linear characteristics by ratio characteristics, eg. length/width ration of parameres, length/width of forceps; or the paramere/forceps measurements could be related to length/width of other part of body reflecting the size of specimen (e.g. width of head capsule, length or width of pronotum).
Comment 3 (to Introduction chapter): Actual number of described Dermaptera species is only about 1900 and number of described Anisolabididae is 393 (Haas 2018). According to comprehensive check-list of Australian earwig, 87 species there are listed (Haas: http://www.earwigs-online.de/AU/au.html) – I recommend to cite this page, as well as the last paper of Fabian Haas (Haas 2018).
Comment 4 (to Introduction/Discussion chapter): Another grain pest species reported from Australia (but not mentioned in the Introduction or Discussion of the MS) is Euborellia annulipes. According to the last research (Kocarek et al. 2015), this species feeds predominantly on plant tissues and it could be a pest of cultivated plants.
References cited in comments:
Haas, F. 2018: Biodiversity of Dermaptera, pp. 315-334. In: Foottit R.G., Adler P.H. (Eds.): Insect Biodiversity: Science and Society II. John Wiley & Sons Ltd.
Kocarek, P., Dvorak, L., & Kirstova, M. (2015). Euborellia annulipes (Dermaptera: Anisolabididae), a new alien earwig in Central European greenhouses: potential pest or beneficial inhabitant?. Applied Entomology and Zoology, 50(2), 201-206.
SAKAI S. 2000: A basic survey for integrated taxonomy of the Dermaptera of the world. Vol. 1. Forficula 4: 1–297.
SRIVASTAVA G. K. 1999: On the higher classification of Anisolabididae (Insecta: Dermaptera) with a check-list of genera and species. Records of Zoological Survey of India 97: 73–100.
SRIVASTAVA G. K. 2003: Dermaptera part II. Superfamily: Anisolaboidea. Fauna of India and adjacent countries. Zoological Survey of India, Kolkata, 235 pp.
Author Response
Response to Reviewer 1
Point 1: “The Anisolabididae have not been yet assessed using numerical or cladistic methods. The taxonomic positions of individual genera are uncertain, as well as the phylogenetic relationships within the family. Therefore the taxonomy of Anisolabididae needs to be consider as provisional and a rigorous phylogenetic analysis of the subfamily based on molecular evidence is necessary to clarifying the relevance and the relationships of the genera that have been described to date. Diagnoses of individual genera are based on length and shape of male parameres, but it is evident, that such system si rather artificial. Anyway, there are more classifications than that published by Steinmann (1989), and evaluated MS dealing with the intergenera relationships needs to contain more precise discussion of this problematics. At least, the most recent classification of Srivastava (1999), used also in his later monograph (Steinmann 2003), should be mentioned and discussed, because according to this classification, based exclusively on male genitals, all species of Australian Gonolabis mentioned in evaluated paper belong to genus Mongolabis (!). Another source, Sakai (2000), is worth mentioning.”
Response: We are particularly grateful for the taxonomic expertise the reviewer brings to their comments. The author responsible for the morphological treatment is currently unable to access the original specimens, and so a proper redress of morphological work is difficult (especially given the time frame). However, the following changes have been made. We hope they are sufficient.
We have included in the methods a qualification of our identifications, mentioning that they should be treated as provisional and mentioned that Srivastava’s classification would yield different identifications (lines 125, 126, 130-136), as well slightly reframed the introduction (lines 98-99) and abstract (lines 21-34) to mention that the purpose of this paper is less systematic and more to do with testing our ability to enumerate Anisolabididae diversity and the utility of the parameres and forceps to this purpose. We have also included in the discussion a paragraph (lines 662-674) discussing the suggestions in Srivastava (1999) and their impact on the results.
Unfortunately, we were unable to source any physical or digital copies of Sakai (2000) or Steinmann (2003).
Point 2: “Earwigs (incl. Anisolabididae) showing high variability in body length, therefore the linear morphometric measures used by the authors are probably correlated with the body length of particular individual (possibly rather than with species). Especially, length and width of the parameres, and maximum length and width of the forceps are cases characters associated with the body robustness. And the body robustness could be associated with local (actual) environmental conditions, esp. unfavourable climatic fluctuations, as low humidity or high temperature. Such conditions could lead to higher ratio of small specimens in the population (not only within, but also across syntopically occurring species). Besides that, forceps are differently curved in males of some Anisolabididae species in relation to their size – small specimens show nearly parallel-sided forceps, bigger specimens hooked. I recommend to test the dependency of measured characters on the body size in studied species. If the lengths are dependent on body size, I suggest replacing all these linear characteristics by ratio characteristics, eg. length/width ration of parameres, length/width of forceps; or the paramere/forceps measurements could be related to length/width of other part of body reflecting the size of specimen (e.g. width of head capsule, length or width of pronotum).”
Response: New data on body size could not be collected as the investigator responsible for this dataset is not in a position to access the original specimens. As a precautionary measure against the effect of dependency of body size, we have reduced our morphometric dataset to four variables: the forceps asymmetry index, the paramere length-width ratio, the basal width of the forceps, and the forceps length-width ratio. The pairwise correlation coefficients for all these were less than 0.46. The methods (lines 138-153) have been updated to reflect this, as well as the results (lines 367-379, Table 1) and Supplementary Tables 2 and 3.
Point 3: “Actual number of described Dermaptera species is only about 1900 and number of described Anisolabididae is 393 (Haas 2018). According to comprehensive check-list of Australian earwig, 87 species there are listed (Haas: http://www.earwigs-online.de/AU/au.html) – I recommend to cite this page, as well as the last paper of Fabian Haas (Haas 2018).”
Response: References to the paper and website suggested by Reviewer 1 have been included (lines 39, 41, 45, 48).
Point 4: “Another grain pest species reported from Australia (but not mentioned in the Introduction or Discussion of the MS) is Euborellia annulipes. According to the last research (Kocarek et al. 2015), this species feeds predominantly on plant tissues and it could be a pest of cultivated plants.”
Response: We have mentioned Euborellia annulipes in the introduction paragraph dealing with potential Australian pest earwigs (lines 63-64), as well as a line in the discussion (lines 640-643) suggesting why it was not present in the sample.
We thank Reviewer 1 for their taking the time to review and comment on our manuscript.

Reviewer 2 Report
This is a well written, interesting paper on the taxonomy and phylogeny of Australian anisolabidid earwigs. Since the taxonomy of this group is very challenging, this study can provide an important basis not only for future studies of the Australian species, but also for those studies in other regions.
However, for further enhancement of the value of this manuscript, I would like to suggest several changes/additions.
<<Major points>>
The most important problem is that researchers who are interested in this group cannot easily determine the morphospecies according to the information provided in this manuscript. To overcome this problem, the authors should provide a key to the morphospecies they studied.
<<Minor points>>
I wrote my specific comments directly on the pdf manuscript file.

Author Response
Response to Reviewer 2
Point 1: “The most important problem is that researchers who are interested in this group cannot easily determine the morphospecies according to the information provided in this manuscript. To overcome this problem, the authors should provide a key to the morphospecies they studied.”
Response: A key for male and females has been included as Appendix A (lines 789-859), and this has been mentioned in the text (lines 427-429).
Point 2: “Not only in male-male competition, but also used in display to females. See Briceno & Eberhard (1995).”
Response: We have included that male forceps are also used to display to females (line 77).
Point 3: “The parameres are paired in the ALL species of earwigs!!! The penis lobes are also paired in Anisolabididae (and other primitive families), but not in Forficulidae, Chelisochidae, Spongiphoridae, Arixeniidae, and Hemimeridae.”
Response: We have removed this from the methods (line 121) such that our classification of Anisolabididae only really relied on the wingless state of adults.
Point 4: “Measurement (6) is difficult to understand. Please add p' and o' to Figure 1 as the width and length measurements of the opposite branch of the forceps. Then, please express meas (6) using o, o', p, p' and q.”
Response: The description of measurements has been re-written to reflect Figure 1 more appropriately, and the description of the forceps asymmetry index has been rewritten for clarity (lines 138-144).
Point 5: “Not the "penultimate abdominal sternite"? In the earwigs, the sternite of the ultimate abdominal segment is largely reduced and vestigial.”
Response: “Ultimate ventral tergite” has been replaced with “penultimate abdominal sternite” (lines 163-164).
Point 6: “As a measure of repeatability, the authors should use "intraclass correlation" rather than ordinary ones. Please see: https://en.wikipedia.org/wiki/Intraclass_correlation”
Response: The repeatability analysis was re-done, calculating the intra-class correlation for each measurement, and this section has been rewritten accordingly (lines 177-178).
Point 7: “I cannot see a tooth in Fig. 1g. Please clearly indicate it by using an arrowhead.”
Response: Red arrows have been added to Figure 1 indicating the location of the dorsoventrally oriented teeth, and the figure caption has been updated to reflect this.
Point 8: “I cannot understand this measure, because the authors measured only right ones of paired parameres.”
Response: “Paramere asymmetry ratio” on Table 1 has been corrected to “paramere length-width ratio.”
Point 9: “(Judging from Figure 2) I think G. nr. gilesi (not G. sp1) can be correctly classified in 100% cases.“
Response: This has not been changed, as this sentence is a report of the results of the classification analysis, and not a subjective measurement of how easy it is to visually distinguish the morphospecies. In any case, the classification analysis has been redone with different variables pursuant to comments made by Reviewers 1 and 3, and the methods (lines 152-153) and results (lines 396-414) have been rewritten to reflect this.
Point 10: “Not dorsoventral?”
Response: “Small dorsal tooth” has been changed to “small dorsoventral tooth” (line 426).
Point 11: “How many female samples do you have that could not be identified.”
Response: No females were left unidentified; two of the morhpospecies did not have female representatives apparent in the sample by barcoding, and so it is possible that females of these two could have been overlooked, but this is unlikely as, as mentioned in the discussion (lines 580-584), no females were found at the sites where males of these morphospecies were found.
Point 12: “Because not all readers are familiar with the geography of Australia, please explicitly indicate "WA" and "SA" in this map, together with the places of the Spencer Gulf, Great Australian Bight, and Adelaide.”
Response: The map, Figure 3, has been altered indicating that the left and right insets fall in Western and South Australia respectively, and labels (a-d) have been added indicating the locations of the Great Australian Bight, Spencer Gulf, Adelaide, and Lake Alexandria.
Point 13: “Because the authors do not use non-Australian representatives of Anisolabididae, the outgroup taxa used in the study are very distant from the ingroup. So, I am not sure that the root position was stable among the trees constructed with different genes/methods. Please mention whether the trees were consistently rooted before pruning the outgroup taxa.”
Response: We have mentioned that trees were rooted by the outgroups prior to pruning (line 302).
Point 4: “I think Gonolabis is paraphyletic, but Anisolabis is POLYphyletic in the trees shown in Figure 4.”
Response: We have mentioned that Gonolabis and Anisolabis appear paraphyletic and polyphyletic respectively (lines 525-526).
Point 15: “Many species of Anisolabididae are flightless. I think this is also an important factor for their endemism. But, this family also includes some cosmopolitan species, living in seasides, such as Euborellia annulipes and Anisolabis maritima.”
Response: We have mentioned that the winglessness of Anisolabididae may contribute to their endemism (lines 639-643) and the exceptions of Euborellia annulipes and Anisolabis maritima.
Point 6: “I agree that some previous workers on the earwig taxonomy relied too much on the paramere morphology for the generic classification. Please refer Kamimura et al. (2016) for a similar criticism. Kamimura, Y., Nishikawa, M. and Lee, C.-Y. (2016) A new earwig of the genus Echinosoma from Penang Island, Peninsular Malaysia, with notes on the taxonomic and nomenclatural problems of the genus Cranopygia (Insecta: Dermaptera: Pygidicranidae). ZooKeys 636: 51-65. doi: 10.3897/zookeys.636.10592”
Response: A sentence with a reference to Kamimura et al. (2016) has been included in the discussion with relation to the discussion of the use of parameres in family-level phylogeny (lines 659-661).
We thank Reviewer 2 for their taking the time to review and comment on our manuscript.

Reviewer 3 Report
The present manuscript „Morphological and molecular analysis of Australian earwigs (Dermaptera) points to unique species and regional endemism in the Anisolabididae family”, submitted by O. P. Stuart and co-authors, submitted to Insects, is a comprehensive and detailed morphological and molecular study on a so far understudied group, Australian Dermaptera: Anisolabididae. The manuscript is well-written and fits into the scopes of the journal. I especially like the combination of morphometrics and molecular analyses. I recommend publishing the article after minor revisions; I have only a few questions that should be addressed.
I suggest to include the year(s) of collecting individuals for the analyses. Authors indicate that they collected during summer and winter at all sites – did you find any differences in the dermapteran diversity between seasons and years and how did you handle this? Did you pool all individuals for the following morphometric and molecular analyses? Since you collected monthly (for 12 months, I suppose?) at a reduced set of sites I wonder if this might give a sampling bias?
I wonder if some measure of overall body size, e.g. leg (tibia?) length, pronotum length/width or similar might be necessary to be included in the PC analysis (did you scale and center variables by morphospecies?). PC1 seems to me associated with body size. Additionally, two of the seven variables are calculated from two other variables used in the same dataset. Such a high correlation might be problematic and should be addressed in the discussion.
After having assigned females to morphospecies by barcoding – did you consider also looking at details of the ovipositor or spermatheca? This might reveal more taxonomic relevant characters than colouration or forceps morphology in females.
The result on haplotype networks (lines 442-447) does not fit into the analysis of Anisolabididae, it only refers to figures in the supplement and there is no discussion for this result. I recommend cutting this part.
Do you need an official permit to collect earwigs in Australia?
Minors: Reference 17 is strange – should some link be included here? “damate” should read “damage” and “gain” “grain”
Reference 49 lacks a Journal name
Author Response
Response to Reviewer 3
Point 1: “I suggest to include the year(s) of collecting individuals for the analyses. Authors indicate that they collected during summer and winter at all sites – did you find any differences in the dermapteran diversity between seasons and years and how did you handle this? Did you pool all individuals for the following morphometric and molecular analyses? Since you collected monthly (for 12 months, I suppose?) at a reduced set of sites I wonder if this might give a sampling bias?”
Response: We have mentioned in the methods the beginning and end dates of collection for the Anisolabididae specimens (lines 110-111).
We have included in the results (lines 429-433) a section describing the inter- and intra-annual variation in collections and have indicated in the methods (lines 134-136) that specimens were pooled by morphospecies across sites and sampling dates. We are very grateful for the reviewer having pointed out seasonal variation; this is relevant to the discussion of Anisolabididae as pests or predators in grain systems. We have therefore also included a paragraph in the discussion briefly addressing this feature of the data (lines 644-649).
We have included in the results (lines 436-438) a paragraph outlining the apparent effect of sampling bias on the relative abundances of the morphospecies in the sample, as well as a sentence in the discussion (lines 586-588) which points out that the relative scarcity of two of the morphospecies in the sample is likely the effect of sampling bias.
Point 2: “I wonder if some measure of overall body size, e.g. leg (tibia?) length, pronotum length/width or similar might be necessary to be included in the PC analysis (did you scale and center variables by morphospecies?). PC1 seems to me associated with body size. Additionally, two of the seven variables are calculated from two other variables used in the same dataset. Such a high correlation might be problematic and should be addressed in the discussion.”
Response: The investigator responsible for collection of morphometric measurements is not in a position to access the original specimens, and so new data cannot be collected, but the morphometric analysis has been redone on a reduced set of variables: the forceps asymmetry index, the paramere length-width ratio, the basal width of the forceps, and the forceps length-width ratio. The index is standardised by the basal width of the forceps, which we used as a rough measure of body robustness (the width of the terminal abdominal segments). Correlations between these four variables were all 0.46 or less. The methods (lines 152-153) and the results (lines 366-384) have been updated to reflect this, as well as Supplementary Tables 2 and 3.
Point 3: “After having assigned females to morphospecies by barcoding – did you consider also looking at details of the ovipositor or spermatheca? This might reveal more taxonomic relevant characters than colouration or forceps morphology in females.”
Response: We did not investigate the spermatheca and ovipositor of unknown females, and the investigator responsible for morphological treatment is currently unable to access original specimens and so new data of this kind cannot be collected. In lieu of a proper treatment of female genitalia, the importance of female genitalia and the suggestion of their future use has been added to the discussion (lines 576-579).
Point 4: “The result on haplotype networks (lines 442-447) does not fit into the analysis of Anisolabididae, it only refers to figures in the supplement and there is no discussion for this result. I recommend cutting this part.”
Response: This section has been removed, along with figures and tables, and figure and table references in the text have been corrected.
Point 5: “Do you need an official permit to collect earwigs in Australia?”
Response: We did not require permits to sample earwigs as all sampling was performed on private land with permission, and we thank the landowners in our acknowledgements (lines 781-785).
Point 6: “Reference 17 is strange – should some link be included here? “damate” should read “damage” and “gain” “grain””
Response: References 17 (now reference 20) has been fixed (line 912). The database is not publicly accessible but is maintained partially by one of the authors (P. Umina).
Point 7: “Reference 49 lacks a Journal name.”
Response: Reference 49 (now reference 53) has been corrected (lines 992-994).
We thank Reviewer 3 for their taking the time to review and comment on our manuscript.

Round 2
Reviewer 1 Report
The authors improved the manuscript according to my comments and I´m fully satisfied with the current version of the article.
Author Response
NA
Reviewer 2 Report
I really appreciate the thorough revision of the manuscript by the authors.
I have only some minor comments on the newly added “key.”
(1) #2
2 Forceps straight, short, no true teeth but the medial surface may be slightly ridged, no parameres present……………………………………………………15 (NOT 14!)
(2) #5, 6, 17, 18
Body length including forceps?
(3)#2
Forcipate should be in italic.
Author Response
Again, we are grateful for the comments on the matter of the key. Following are our corrections pursuant to Reviewer 2’s comments.
Comment 1: “#2 Forceps straight, short, no true teeth but the medial surface may be slightly ridged, no parameres present……………………………………………………15 (NOT 14!)”
Response: We have corrected this error (line 798).
Comment 2: “#5, 6, 17, 18 Body length including forceps?”
Response: It has been included that the measurements given include forceps length (lines 807-808, 810-811, 847, 850) and also have added a brief note to steps #5 and #6 on differentiating the male forceps of the South Australian and Western Australian clades (lines 807-808, 810-811).
Comment 3: “#20 Forcipata should be in italic.”
Response: We have corrected this error (line 857).
Reviewer 3 Report
The revision of the manuscript „Morphological and molecular analysis of Australian earwigs (Dermaptera) points to unique species and regional endemism in the Anisolabididae family”, submitted by O. P. Stuart and co-authors seems well done to me; all questions raised were thoroughly answered. I recommend accepting the manuscript, however, I have one concern that needs to be addressed before: Table S2 now gives the ratios of forceps length / width. If re-calculating these with the mean values given in the table, I do not get these results, additionally, minimum values are sometimes given as 0.00, which could not really be possible for a ratio of two lengths… for G. sp.1, the max. range is smaller than the mean value… can you please make sure that these ratios have been calculated correctly?
Author Response
We are grateful that Reviewer 3 has taken the time to check over the supplementary material. The confusion over our summary statistics led us to re-examine our morphometric analysis, and we encountered an error in our calculation such that the ‘forceps ratio’ was calculated as the absolute difference between maximum length and width, and not their ratio.
We have taken the following steps to correct this:
The summary statistics have been recalculated (Table S2). We point out that the means values for the original variables cannot be used to recreate the mean values for the derived variables. For example, the forceps ratio was calculated for each individual, and the mean value presented in Table S2 is the mean of these values. The mean values for the two original variables (maximum forceps length/width) are not used to calculate the mean derived variable. If all original variables were perfectly normally distributed, then the means calculated both ways would be the same, but of course the variables do not follow a perfect normal distribution. That said, the ratio of the mean maximum forceps length and width should approximate the mean forceps ratio, and welcome further checking.
The PCA has been redone and results updated (Figure 2, Table 1, lines 371-385).
The classification analysis has been redone and results updated (Table S3, lines 401-418).
We hope these changes are satisfactory. Broadly, the results do not change, but the specifics have been adjusted.
Round 3
Reviewer 3 Report
Thanks for correcting Table S2, this seems fine to me now.